# The critical role of membralin in postnatal motor neuron survival and disease

Bo Yang[1][†][‡], Mingliang Qu[1][†][§], Rengang Wang[1], Jon E Chatterton[1], Xiao-Bo Liu[2], Bing Zhu[1], Sonoko Narisawa[3], Jose Luis Millan[3], Nobuki Nakanishi[1], Kathryn Swoboda[4], Stuart A Lipton[1,5], Dongxian Zhang[1]*

[1]Neuroscience and Aging Research Center, Sanford-Burnham Medical Research Institute, La Jolla, United States; [2]Electron Microscopy Laboratory, Department of Pathology and Laboratory Medicine, School of Medicine, University of California, Davis, Davis, United States; [3]Sanford Children's Health Research Center, Sanford-Burnham Medical Research Institute, La Jolla, United States; [4]Department of Neurology, Massachusetts General Hospital, Boston, United States; [5]Department of Neuroscience, School of Medicine, University of California, San Diego, La Jolla, United States

*For correspondence: dzhang@sbmri.org

[†]These authors contributed equally to this work

Present address: [‡]Eugenom, Inc., San Diego, United States; [§]Shanghai Yuanqi Clinical Lab Ltd., Shanghai, China

Competing interests: The authors declare that no competing interests exist.

**Abstract** Hitherto, membralin has been a protein of unknown function. Here, we show that membralin mutant mice manifest a severe and early-onset motor neuron disease in an autosomal recessive manner, dying by postnatal day 5–6. Selective death of lower motor neurons, including those innervating the limbs, intercostal muscles, and diaphragm, is predominantly responsible for this fatal phenotype. Neural expression of a membralin transgene completely rescues membralin mutant mice. Mechanistically, we show that membralin interacts with Erlin2, an endoplasmic reticulum (ER) membrane protein that is located in lipid rafts and known to be important in ER-associated protein degradation (ERAD). Accordingly, the degradation rate of ERAD substrates is attenuated in cells lacking membralin. Membralin mutations or deficiency in mouse models induces ER stress, rendering neurons more vulnerable to cell death. Our study reveals a critical role of membralin in motor neuron survival and suggests a novel mechanism for early-onset motor neuron disease.

## Introduction

The endoplasmic reticulum (ER) is a membrane-enclosed cellular organelle that plays an essential role in the folding of membrane-bound and secreted proteins, synthesis of lipids and sterols, and storage of free $Ca^{2+}$. ER stress is often triggered by the accumulation of unfolded proteins due to pathological conditions, such as DNA sequence mutations, transcriptional and translational errors, or protein folding failure (*Kim et al., 2008*; *Lin et al., 2008*). Mammalian cells have evolved an intricate system with multiple signaling pathways to respond to ER stress, collectively termed the unfolded protein response (UPR) (*Kim et al., 2008*; *Lin et al., 2008*; *Walter and Ron, 2011*). If unchecked, ER stress eventually causes cell death, including neuronal death in neurodegenerative diseases (*Lindholm et al., 2006*; *Scheper and Hoozemans, 2009*; *Hetz and Mollereau, 2014*).

Increased ER stress is thought to play an early role in motor neuron diseases, including amyotrophic lateral sclerosis (ALS) and Charcot-Marie-Tooth (CMT) (*Atkin et al., 2006*; *Nagata et al., 2007*; *Nishitoh et al., 2008*; *Kanekura et al., 2009*; *Saxena et al., 2009*). Manifestations and consequences of ER stress have been studied in the pathogenesis of SOD1 mutant mice, a commonly used model of ALS that expresses mutant human SOD1 protein as found in some inherited forms of ALS (*Atkin et al., 2006*; *Kanekura et al., 2009*; *Saxena et al., 2009*). Many ER stress-related molecules are upregulated in SOD1 mice at an early stage of the disease, and some are specific to motor neurons (*Tobisawa et al., 2003*; *Atkin et al., 2006*; *Kikuchi et al., 2006*; *Nagata et al., 2007*;

**eLife digest** As new proteins are built inside a cell, many will pass into a structure called the endoplasmic reticulum for processing. There, the proteins are folded into the specific three-dimensional shapes that allow them to carry out their respective jobs. Sometimes the folding process goes awry, leading to a build-up of unfolded proteins that stress the endoplasmic reticulum and can kill the cell. Brain cells are particularly vulnerable to death from endoplasmic reticulum stress. To combat a deadly build-up of unfolded proteins, each cell has systems that respond when the endoplasmic reticulum is under stress.

Unchecked stress on the endoplasmic reticulum has been linked to diseases like amyotrophic lateral sclerosis (called ALS for short). In diseases like ALS, the nerve cells that control muscle movements gradually die off, causing a loss of muscle control and eventually death. Scientists suspect that these nerve cells (called motor neurons) are particularly sensitive to endoplasmic reticulum stress because they are highly active. Drugs that help counteract stress on the endoplasmic reticulum extend the lives of mice with motor neuron disease, suggesting this may be a useful strategy for treating such diseases in humans.

Now, Yang, Qu et al. identify a new protein that appears necessary for a healthy endoplasmic reticulum. Mice that lack the gene for a protein called membralin die within five or six days after birth because their motor neurons die off. Further experiments showed that re-introducing membralin in their nervous system can rescue these membralin-deficient mice.

Yang, Qu et al. found that membralin interacts with another protein that helps eliminate poorly folded or unfolded proteins in the endoplasmic reticulum, and thus relieves stress on the cell. Mutations in this endoplasmic reticulum stress response protein have previously been linked to motor neuron diseases. The motor neurons in membralin-deficient mice show signs of endoplasmic reticulum stress and are extra vulnerable to chemicals that induce protein misfolding. Together, the experiments show membralin plays an important role in mitigating stress on the endoplasmic reticulum. More studies of mice lacking membralin may help explain why the endoplasmic reticulum stress increases in motor neuron diseases and may point to possible treatments.

---

*Ito et al., 2009*; *Saxena et al., 2009*). In addition, reduction in the ER co-chaperone SIL1 has been found to be associated specifically with ER stress-prone, fast-fatigable motor neurons (*Filezac de L'Etang et al., 2015*). ER upregulation of the UPR and aberrant modification of protein disulfide isomerase (PDI) also occur in human tissues from sporadic ALS (*Atkin et al., 2008*; *Walker et al., 2010*). Our studies on the activation of the type I interferon signaling in astrocytes at pre-symptomatic stages of SOD1 mice also suggest that ER stress in motor neurons may play a crucial role in disease onset (*Wang et al., 2011*). Moreover, genetic interruption of UPR signaling molecules, such as XBP-1 or apoptosis signal-regulating kinase1 (ASK1), protects motor neurons in SOD1 mutant mice (*Nishitoh et al., 2008*; *Hetz and Mollereau, 2014*). It has been suggested that the unusually heavy metabolic demand of motor neurons may make them particularly susceptible to ER stress (*Vinay et al., 2000*; *Carrascal et al., 2005*; *Li et al., 2005*). Importantly, ER stress inhibitors, such as salubrinal, guanabenz, and sphin1, delay disease onset and prolong the survival of these mutant mice (*Saxena et al., 2009*; *Jiang et al., 2014*; *Wang et al., 2014*; *Das et al., 2015*). This finding suggests that ER stress-related molecules may represent rational drug targets for motor neuron diseases.

Previously, membralin had been predicted by genome sequencing to encompass an open reading frame (*orf61* in mouse and *c19orf6* in human), but the function of the encoded protein has, heretofore, remained unknown. *Orf61* is highly conserved and found in the genome of most species except yeast and bacteria. cDNAs for membralin have been cloned from human and mouse tissues (*Andersson and von Euler, 2002*; *Chen et al., 2005*), and the protein encoded by *orf61* was named membralin because it was predicted to be a membrane protein (*Andersson and von Euler, 2002*). Given the absence of known protein domains, membralin may represent a novel class of proteins.

Here, we discovered during our cloning and characterization of the glutamate receptor subunit 3B gene (GluN3B, formerly designated NR3B, *Nishi et al., 2001*; *Chatterton et al., 2002*; *Matsuda et al., 2002*) that the 3′ end of the GluN3B gene overlaps with the 3′ end of the membralin gene on the opposite strand. Thus, our GluN3B knockout (KO) mice also carry a C-terminal truncation of

membralin. We observed that GluN3B/membralin C-terminal double-knockout (DKO) mice die at postnatal day 5–6 of paresis due to severe motor neuron degeneration. Transgenic expression of membralin, but not GluN3B, rescued the phenotypes of DKO mice. Additionally, we generated membralin-specific KO mice and found that they phenocopied the DKO mice. These data suggest that membralin plays a critical role in motor neuron survival. We also demonstrate that membralin interacts with Erlin2, a protein that is enriched in ER lipid rafts and important for ER-associated protein degradation (ERAD) (*Ikegawa et al., 1999*; *Browman et al., 2006*). Additionally, neurons from membralin mutant mice display increased basal ER stress and increased vulnerability to ER stress-inducing agents. Our discovery that membralin mutations result in motor neuron disease provides mechanistic insight into pathophysiology and may offer a novel target for therapy.

## Results

### Membralin KO mice manifest severe motor neuron loss

We generated GluN3B-deficienct mice by deletion of the 8-kb region encoding the entire GluN3B gene and found that it also resulted in partial deletion of membralin (*Figure 1—figure supplement 1*). Sequence analysis confirmed that the translation frame of the altered membralin transcript in null mice terminates 147 nucleotides (nts) beyond the end of partial exon XI, resulting in a slightly truncated membralin protein with 49 unrelated amino-acid (aa) residues replacing the wild-type (WT) 89 C-terminal aa residues. Thus, our targeting strategy generated a GluN3B/membralin C-terminal DKO mouse line.

DKO mice appeared normal at birth but unexpectedly died at postnatal day 5–6 (P5-6). The motor function of these mice was indistinguishable from WT or heterozygous littermates during the first three postnatal days, but motor strength was severely impaired, thereafter, preceding death. DKO mice were also significantly smaller than WT and heterozygous littermates (*Figure 1A*) but displayed normal gross anatomy and histology in the brain (*Figure 1—figure supplement 2*), visceral organs, and skeletal muscle.

As DKO mice showed signs of paresis, we conducted histological examination of motor neurons in the lumbar spinal cord using the motor neuron-specific marker, Hb9. The number of motor neurons in DKO mice was unchanged at P0 and P3, but reduced by 40% at P5, compared to littermate WT controls (*Figure 1B,C*). The remaining motor neurons appeared smaller than those of WT mice. Motor neuron loss and surrounding astrogliosis were also apparent in the cervical spinal cords of P5 DKO mice (*Figure 1D*). We next examined Isg15 expression (interferon-induced 17 kDa protein), which is thought to play a role in innate immunity and is upregulated at both pre-symptomatic stages and post-symptomatic stages in a mouse model of ALS (*Wang et al., 2011*). WT mice manifested minimal Isg15 expression throughout the brain and spinal cord. In contrast, by P3, DKO mice had increased Isg15 expression in the ventral horn of the cervical and lumbar enlargement of the spinal cord (*Figure 1—figure supplement 3A*). By P5, DKO mice showed increased Isg15 expression in additional areas containing primary motor neurons, with highest expression in the ventral horn of all segments of the spinal cord and moderate expression in the facial nucleus of the brainstem (*Figure 1—figure supplement 3A,B*). No Isg15 expression was detectable in other brain areas, even in late-stage DKO mice. These results suggest that the motor neurons are progressively injured in DKO mice, starting with those in the spinal cord. Since the normal process of programmed cell death in embryonic motor neurons is complete at birth (*Lance-Jones, 1982*), the loss of motor neurons in DKO mice was most likely due to cellular events occurring perinatally.

To confirm specific loss of motor neurons in DKO mice, we examined the spinal roots of the lumbar spinal cord by electron microscopy. Motor neuron axons in the ventral root of DKO mice showed advanced degeneration (*Figure 1E*), whereas sensory neuron axons in the dorsal root were intact (*Figure 1F*). Under high magnification of motor neuron axons, we saw intra-axonal vacuoles and the loss of axoplasm (*Figure 1G*), as well as disintegrated myelin and amorphous lipids (*Figure 1H*), as typically seen in Wallerian degeneration. The percentage of degenerated axons in the ventral root of DKO mice reached $48.0 \pm 4.1\%$ by P5, which was significantly greater ($p < 0.01$) than in WT mice ($8.6 \pm 1.5\%$) (*Figure 1I*). In contrast, the percentage of degenerated axons in the dorsal root of DKO mice ($10.5 \pm 1.9\%$) was similar to that of the WT ventral root, consistent with the normal background level of axonal degeneration that occurs during development. Moreover, the

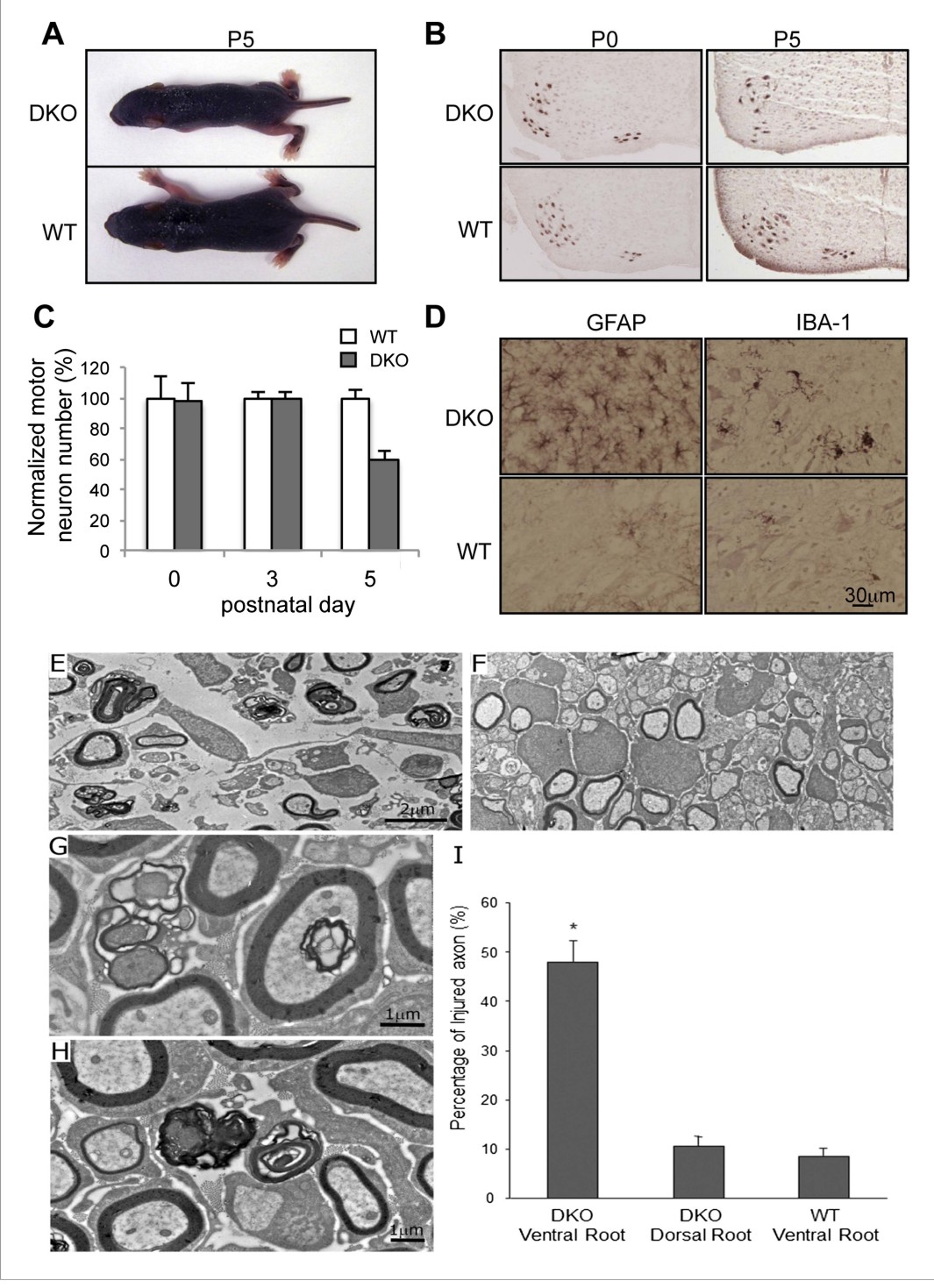

**Figure 1**. Motor neuron death in GluN3B/Membralin DKO mice. (**A**) DKO and WT littermate mice are shown at P5.
The DKO mouse was unable to maintain correct posture. (**B**) Immunostaining with anti-Hb9 antibody revealed
a dramatic decrease in the number of motor neurons in the lumbar spinal cord of DKO mice relative to WT littermate
mice at P5 but not at P0. (**C**) The number of motor neurons in the lumbar spinal cord of KO and WT mice was not
significantly different at P0 and P3. However, severe motor neuron loss was observed by P5 in DKO mice ($n = 3$)
compared with WT mice ($n = 3$, *$p < 0.05$ by Student's $t$-test). (**D**) Glial fibrillary-associated protein (GFAP) and
ionized calcium-binding adapter molecule 1 (IBA-1) immunostaining in the spinal cord of WT and DKO mice.
GFAP-positive astrocytes and IBA-1-positive microglial cells surrounding motor neurons (counterstained with
Giemsa) of DKO mice showed swelling cell bodies and processes, a typical sign of astrogliosis. (**E**,**F**) Electron
*Figure 1. continued on next page*

*Figure 1. Continued*

microscopic analysis of spinal roots showed that DKO mice at age P5 had severe damage of motor neuron axons in the ventral root (**E**) but not of sensory neuron axons in the dorsal root (**F**). Bar, 2 μm. (**G,H**) Damaged axons in the ventral root showed either vacuoles or loss of axoplasm (**G**), as an early sign of axonal damage. At a later stage of axonal damage, dark disintegrated myelin sheaths and amorphous lipid were present (**H**). Bar, 1 μm. (**I**) Percentage of injured axons in the ventral and dorsal roots of DKO and WT mice (*p < 0.05 by Student's *t*-test, n = 3). Data are mean ± s.e.m.

The following figure supplements are available for figure 1:

**Figure supplement 1**. Generation of GluN3B/membralin DKO mice.

**Figure supplement 2**. GluN3B/membralin DKO and WT mice show similar gross anatomy of the brain.

**Figure supplement 3**. Expression of the motor neuron injury marker ISG15 in the spinal cord and brainstem of membralin KO mice.

**Figure supplement 4**. Degeneration of motor neuron fibers in phrenic nerve of GluN3B/membralin DKO mice.

sensory afferent terminals in the spinal cord of DKO mice appeared normal (*Figure 1—figure supplement 4A*). In contrast, we found degeneration of phrenic motor nerve fibers and terminals in P5 DKO mice (*Figure 1—figure supplement 4B*). These results further confirm the selective death of motor neurons in DKO mice, potentially leading to respiratory failure due to loss of motor neuron innervation of the intercostal muscles and diaphragm. Taken together, our DKO mice present a phenotype of early-onset and apparently selective motor neuron degeneration.

## Causality of membralin deletion in motor neuron degeneration

Since both membralin and GluN3B are mutated in DKO mice, we needed to determine which gene was responsible for the motor neuron degeneration. Therefore, we tested whether transgenically expressed intact membralin or GluN3B could rescue DKO mice. We generated a transgenic mouse line carrying full-length membralin under the control of the murine *prion* promoter, followed by internal ribosome entry site (IRES) and enhanced green fluorescent protein (EGFP) cDNA sequences (*Figure 2A*). The membralin transgenic mice [Tg (membralin)] were fertile, normal in size, and did not display any gross physical or behavioral abnormalities compared with their littermate controls. When Tg (membralin) mice were crossbred with DKO mice, the DKO/Tg (membralin) mice were apparently normal, manifesting neither paresis nor premature death (*Figure 2B*). Thus, the membralin transgene rescued DKO mice, even though membralin mRNA levels remained somewhat lower than in WT mice (*Figure 2C*). GFP immunostaining in the ventral horn of the spinal cord showed that membralin/GFP transgene expression was predominantly neuronal, although we could not rule out relatively weak expression in glial cells (*Figure 2—figure supplement 1*). The DKO/Tg (membralin) mice showed no sign of motor defects or motor neuron loss, despite the absence of GluN3B expression (*Figure 2D*). In contrast, the GluN3B transgene did not rescue DKO mice (*Figure 2—figure supplement 2*), and GluN3B KO mice generated by deleting the first exon did not show any defect in motor neurons (*Niemann et al., 2007*). Taken together, these findings are consistent with the hypothesis that the loss of intact membralin in DKO mice causes motor neuron degeneration.

Since heterozygous DKO mice did not manifest motor defects, and expression of the truncated form of membralin appears to lack any deleterious effect. This conclusion is also supported by results from a second membralin KO mouse line we generated using a gene trapping strategy (*Figure 3A*). RT-PCR experiments using primers from exon 1 and 2 further confirmed that homozygous mice carrying membralin/trapping vector alleles did not express membralin mRNAs with sequences beyond exon 2. Membralin protein was also reduced in heterozygotes and absent in homozygotes. These membralin KO mice phenocopied the DKO mice in motor defects, with ~50% motor neuron loss preceding death around P5 (*Figure 3B,C*).

## Membralin resides in the ER membrane and interacts with Erlin2

Membralin is a novel protein with no conserved domains. RT-PCR analysis showed that membralin mRNA was expressed in multiple tissues, including brain and spinal cord, but not heart, skeletal

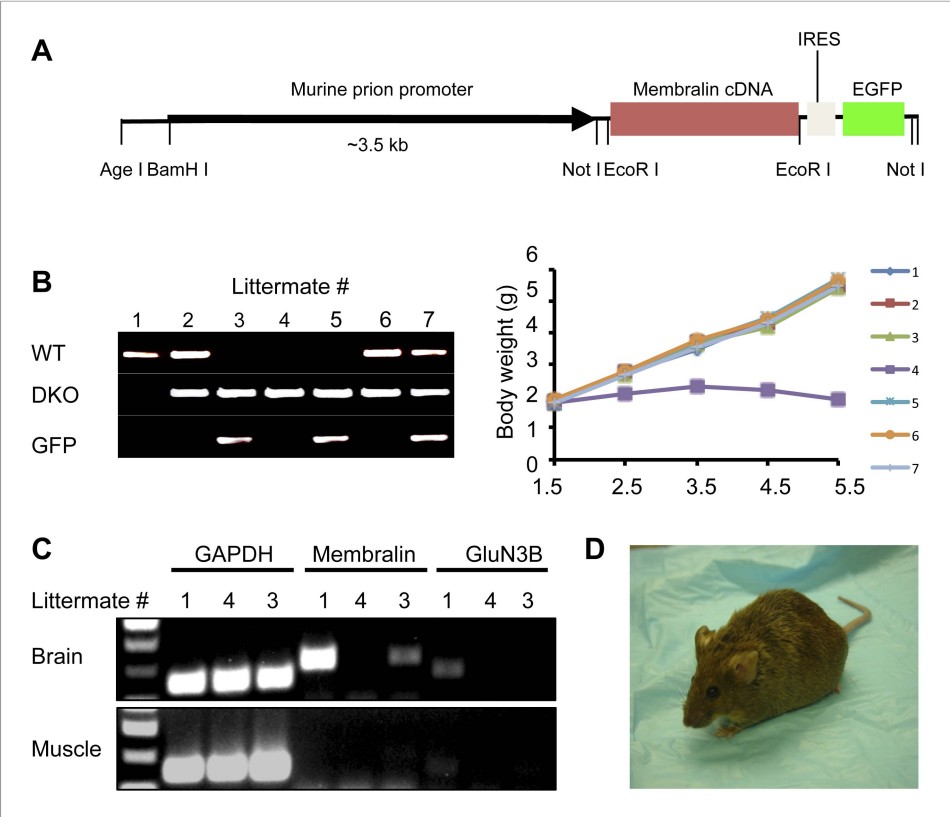

**Figure 2**. Membralin transgene [Tg (membralin)] rescues GluN3B/membralin (DKO) mice. (**A**) Generation of the transgenic constructs expressing membralin. The murine prion promoter was cloned with full-length mouse membralin cDNA followed by the IRES and the EGFP sequence. The entire transgenic sequence was isolated by enzymatic digestion and used to generate Tg (membralin) mice. (**B**) DKO/Tg (membralin) mice were viable and fertile. Left: genotyping by PCR showed that 3 littermates (#3, 4 and 5) from a breeding of DKO and Tg (membralin) mice were positive to DKO primers and negative to WT primers (same primers used as in *Figure 1*). Of the three DKO mice, the two receiving membralin transgene (#3 and 5, as positively identified by GFP primers) were rescued. Right: Weight of the 7 littermates was monitored after birth. Only DKO mouse #4, which was negative for the membralin transgene, showed weight loss after P3 and died at P5.5. Data shown here are from one representative litter (5 independent litters were analyzed with similar results). (**C**) Gene expression analysis by RT-PCR for mice #1, 3, 4 of the same litter as in **B**. Membralin transgene expression is seen in the brains of littermates #1 and 3, but not 4, whereas GluN3B is only expressed in #1. Membralin and GluN3B are not expressed in muscle samples. GAPDH is expressed in both brain and muscles of all mice and served as a positive control. (**D**) Rescued DKO/Tg (membralin) mice lived to adulthood without any sign of paresis.

The following figure supplements are available for figure 2:

**Figure supplement 1**. Expression of GFP-membralin transgene in spinal cord of membralin transgenic mice.

**Figure supplement 2**. GluN3B transgene cannot rescue GluN3B/membralin DKO mice.

muscle, or spleen (*Figure 4A*). Western blot analysis of spinal cord samples from WT and KO mice showed specific expression of membralin in the ER but not in the cytosol or mitochondria (*Figure 4B*). Since our anti-membralin antibody proved ineffective for immunostaining, we made a fusion protein by adding a myc tag to the C-terminus of membralin (membralin-myc). Immunocytochemistry demonstrated that membralin-myc co-localized with an ER marker PDI but not with markers for Golgi or mitochondria (*Figure 4C*). Membralin-myc was not expressed on the cell surface, since it was not detected in transfected cells without first permeabilizing the membrane (data not shown). Software for topology prediction from the Center for Biological Sequences (CBS, TMHMM) indicated that the membralin is most likely a transmembrane protein with 4–6 membrane-spanning regions, containing

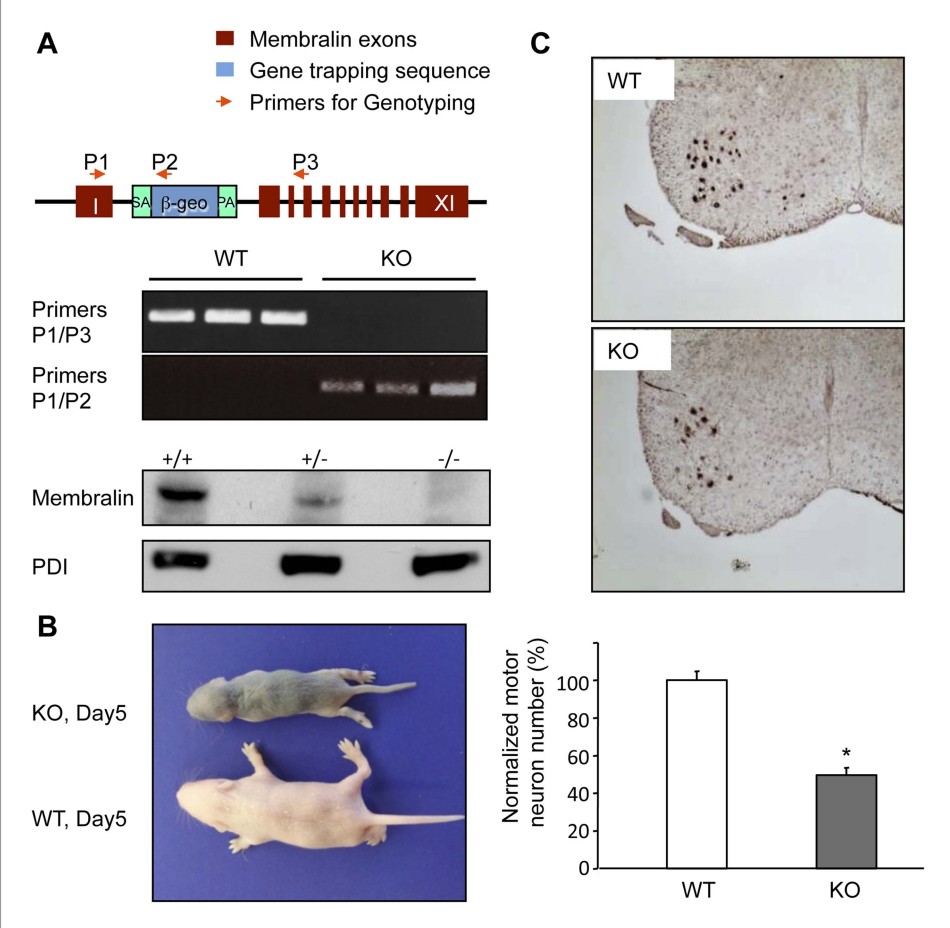

**Figure 3**. Membralin KO mice die of motor neuron degeneration and consequent paresis. (**A**) Gene trapping was used to generate membralin KO mice by inserting a trapping vector that contained a splicing acceptor sequence between exon 1 and 2 to disrupt normal RNA splicing. Primers (P1, P2, P3, red arrows) were designed to detect normal and trapped membralin transcripts. RT-PCR experiments showed that PCR products using the P1 and P3 primer pair were only detected in WT mouse brain (lane 1), liver (lane 2), and kidney (lane 3), whereas PCR products using the P1 and P2 primers were only detected in these tissues of KO mice. (**B**) Membralin KO mice phenocopied GluN3B/membralin DKO mice and died of paresis around P5. (**C**) Lumbar motor neurons, identified by anti-Hb9 staining (top panels), were significantly reduced (lower panel) in membralin KO mice compared to WT mice ($n = 3$ for each group of mice, *$p < 0.05$, Student's $t$-test). Data are mean +s.e.m.

an ER-lumen domain between membrane-spanning regions 1 and 2. Additionally, the protein has multiple glycosylation sites and cytosolic regions at both the N-terminal ends and C-terminal ends (*Figure 4—figure supplement 1A,B*). These data suggest that membralin is a membrane protein most likely localized to the ER and/or outer nuclear envelope.

A recent study defining human ERAD networks through an integrative mapping strategy identified a potential interaction of membralin (C19orf6) with Erlin2 (*Christianson et al., 2012*). Erlin2 is a protein that is enriched in ER lipid rafts (*Ikegawa et al., 1999*; *Browman et al., 2006*), and mutations in Erlin2 have been linked to several human diseases with motor dysfunction (*Al-Yahyaee et al., 2006*; *Alazami et al., 2011*; *Yildirim et al., 2011*; *Al-Saif et al., 2012*; *Wakil et al., 2013*). We designed experiments to verify the membralin/Erlin2 interaction. We found that GFP-tagged membralin not only co-localized with but also manifested a strong FRET signal with mCherry-tagged Erlin2 in the ER (*Figure 5A*). The FRET efficiency was significantly higher for GFP-tagged membralin and mCherry-tagged Erlin2 ($18.00 \pm 0.70\%$) than the negative control (GFP and mCherry, $0.15 \pm 0.05\%$), consistent with a direct interaction of the two molecules (*Figure 5B*). Additionally, Myc-tagged full-length membralin co-immunoprecipitated with HA-tagged Erlin2

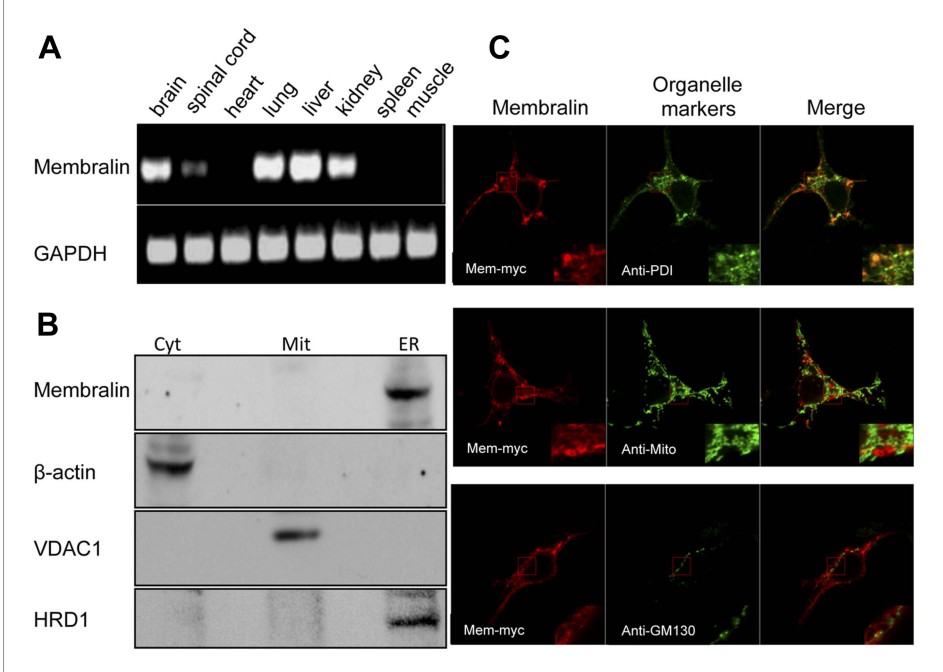

**Figure 4**. Cellular expression and localization of membralin. (**A**) Semi-quantitative RT-PCR analysis showed the expression of membralin mRNA in brain (1), spinal cord (2), lung (4), liver (5), and kidney (6), but not in heart (3), spleen (7), and muscle (8) tissues of WT mice. (**B**) Western blot analysis showed the expression of membralin protein in the ER fraction, but not in the mitochondrial fraction (Mit) or the cytosolic fraction (Cyt), of the WT mouse brains. (**C**) Subcellular localization of Myc-tagged membralin in HEK 293 cells. Cells were transiently transfected with Myc-tagged membralin. At 48 hr after transfection, localization of membralin was detected by confocal microscopy (Zeiss 710). Organelles were labeled with specific markers (anti-PDI for ER, anti-mitochondria for mitochondria, and anti-GMP130 for Golgi). Region of interest shown at higher magnification in insets (bottom right corner). Merged pictures show membralin co-localized with the ER marker, PDI.

The following figure supplement is available for figure 4:

**Figure supplement 1**. Structural prediction of membralin.

(*Figure 5C*). This interaction of membralin with Erlin2 further supports the notion that membralin is an ER membrane protein and also suggests that loss of membralin could potentially increase ER stress by interrupting ERAD. Indeed, we found that the ER membrane protein, CD3-δ, was cleared more slowly in mouse embryonic fibroblasts (MEFs) prepared from membralin KO mice than from WT mice (*Figure 5D*), whereas there was no change in the clearance of the ER lumenal protein, NHK (*Figure 5—figure supplement 1*). These results suggest that membralin deficiency affects degradation of ER membrane proteins, which could potentially increase ER stress. Thus, we next analyzed ER stress in membralin KO mice.

## Increased ER stress in membralin KO mice

Increased ER stress is thought to be involved in some forms of motor neuron disease, including ALS (*Atkin et al., 2006*; *Nagata et al., 2007*; *Nishitoh et al., 2008*; *Kanekura et al., 2009*; *Saxena et al., 2009*). We, therefore, tested whether the absence of membralin impacted ER stress (*Walter and Ron, 2011*). Western blot analysis showed that the levels of GRP78 and ATF4, two molecules induced during ER stress via activation of the PERK signaling pathway, were consistently higher in the spinal cord of P3 membralin KO mice than WT mice ($154 \pm 35\%$ and $412 \pm 39\%$ of WT, respectively; KO, $n = 4$; WT, $n = 3$; *Figure 6A*). In contrast, levels of spliced XBP-1, indicating activation of the IRE1 signaling pathway during ER stress, were not altered (*Figure 6—figure supplement 1A*). Furthermore, tunicamycin, an inducer of ER stress, produced a greater degree of cell death in

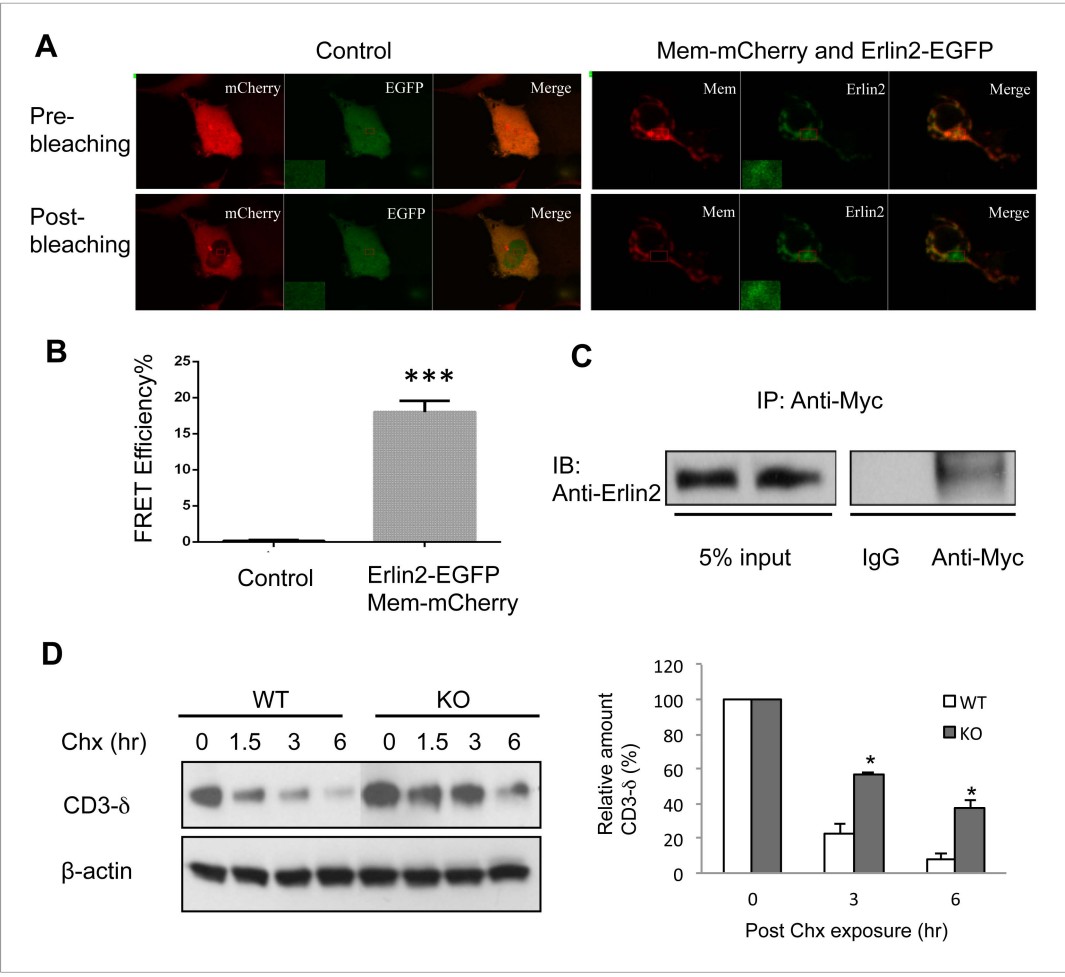

**Figure 5.** Membralin interacts with Erlin2 and regulates protein degradation. (**A**) HEK 293T cells were transfected with control (mCherry and EGFP) or target (membralin-mCherry and Erlin2-EGFP) molecules. Individual or merged fluorescence images were shown in pre-bleaching and post-bleaching conditions. (**B**) FRET efficiency was quantified for control and target groups. All data shown are mean ± s.e.m. n = 10; ***p < 0.001 by Student's *t*-test. (**C**) The interaction of membralin with Erlin2 was confirmed by co-immunoprecipitation experiments using HA-tagged Erlin2 and Myc-tagged membralin fusion proteins co-transfected into HEK 293T cells. Whole-cell lysates were immunoprecipitated with anti-Myc or IgG (negative control) and immunoblotted using anti-Erlin2. (**D**) MEFs from both WT and membralin KO mouse were transfected with HA tagged CD3-δ and subjected to pulse-chase analysis of CD3-δ degradation after exposing with cycloheximide (Chx). Proteins collected at the indicated time points were subjected to immunoblotting with antibodies to HA. Image shown represents the example of immunoblotting and graphs, the quantification of three experiments. (n = 3, *p < 0.05 by Student's *t*-test). All data shown are mean ± s.e.m.

The following figure supplement is available for figure 5:

**Figure supplement 1**. Deletion of membralin did not prolong the half life of ERAD substrate NHK.

MEFs from KO mice than from WT mice (*Figure 6B*). Tunicamycin exposure also increased CHOP and ATF4 levels earlier and to a greater extent in MEFs from KO mice; GRP78 levels were also higher at 24 hr post-exposure (*Figure 6C*). Basal levels of ATF4, but not CHOP, were higher in membralin KO mice than WT mice (*Figure 6C*), but XBP-1 splicing was not altered (*Figure 6—figure supplement 1B*). Collectively, these data suggest that loss of membralin increases basal ER stress and makes cells more vulnerable to additional ER stress-induced injury.

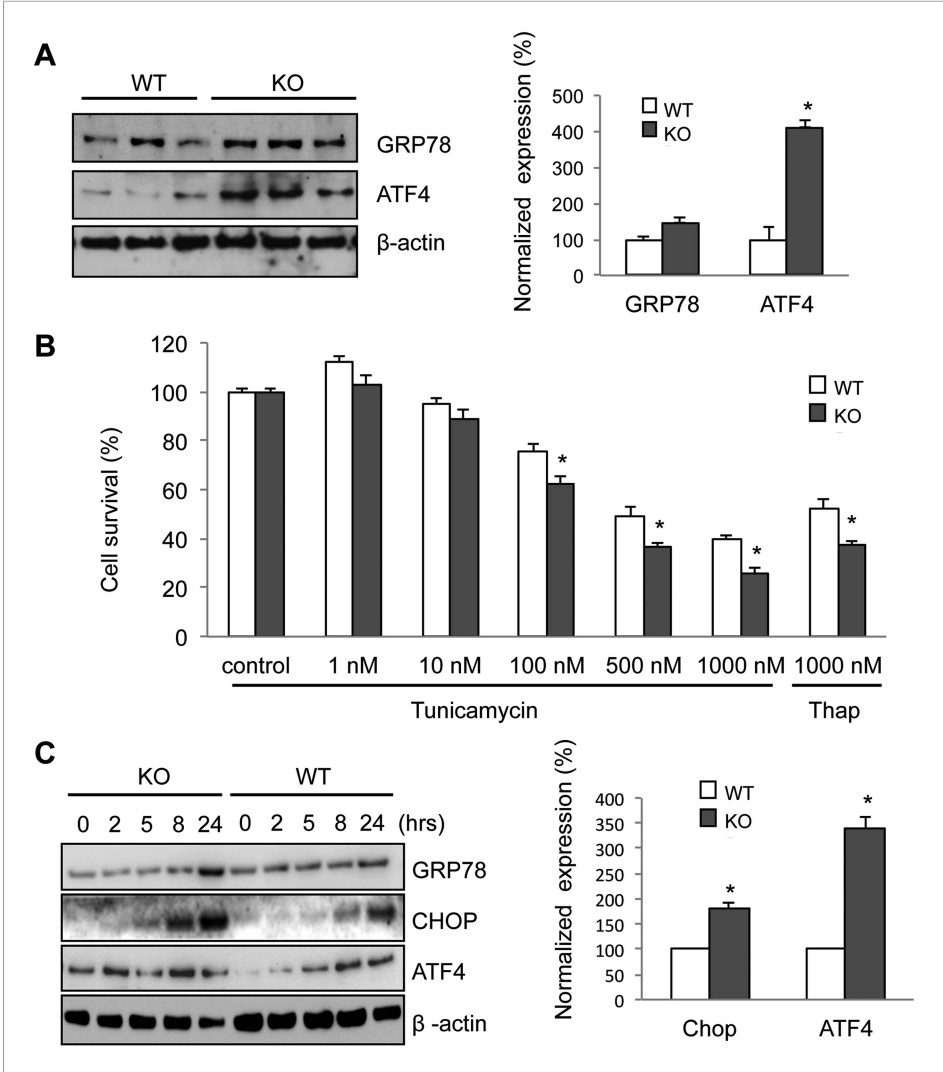

**Figure 6**. Elevated ER stress in membralin KO mice. (**A**) Upregulation of GRP78 and ATF4 in spinal cord of membralin KO mice. Left: immunoblot analysis shows higher levels of GRP78 and ATF4 in the spinal cord of P3 membralin KO mice compared to that of littermate WT mice. Right: increased expression level of GRP78 and ATF4 normalized to actin in membralin KO mice over WT mice. (**B**) Survival of MEFs after 24-hr exposure to tunicamycin or thapsigargin determined by a cytotoxicity assay. (**C**) The expression level of GRP78, CHOP, and ATF4 in MEFs from KO and WT mice is shown after exposure to tunicamycin at different time points. Levels of CHOP and ATF4 are elevated earlier and higher in MEFs from KO than that from WT mice. A representative immunoblot is shown; graphs include data from three experiments. For each panel, data are mean ± s.e.m.; $n = 3$ for each bar; *$p < 0.05$ by Student's $t$-test.

The following figure supplement is available for figure 6:

**Figure supplement 1**. Membralin deletion does not alter Xbp-1 splicing.

## Discussion

Hereditary motor neuropathy (HMN) is usually subdivided into two groups: proximal HMN, i.e., classical spinal muscular atrophy (SMA) syndromes, and distal HMN, which clinically resemble CMT syndromes without obvious sensory abnormalities. In contrast to proximal HMN, distal HMN represents a heterogeneous group of peripheral neuropathies affecting mainly distal muscles with both autosomal dominant and recessive inheritance (*Irobi et al., 2004*, *2006*; *Irobi-Devolder, 2008*). Seven subgroups of distal HMN were initially proposed based on age of onset, mode of inheritance,

and clinical features of a limited number of affected families (*Harding, 1993*). Additional forms of distal HMN were subsequently reported, adding more genetic complexity. A dozen of causal gene loci and other associated genes have been identified for distal HMN (exclusive of type I); these genes encode for a functionally heterogeneous group of proteins, as summarized in recent reviews (*Irobi et al., 2004*, *2006*; *Irobi-Devolder, 2008*). The biological functions of the affected proteins include stress responses (small heat shock proteins HSP22, HSP27), housekeeping (GARS, glycyl tRNA synthetase), protein glycosylation in the ER (seipin), RNA processing (immunoglobulin μ-binding protein 2 [LGHMBP2] and senataxin), and axonal transport (dynactin). It is not clear, however, how these mutated proteins of diverse cellular function converge to affect motor neuron survival selectively. In fact, their expression is not even restricted to motor neurons. A similar case is found for membralin in this study. Therefore, further studies on molecular mechanisms underlying the selectivity of motor neuron death in distal HMN will be critical to understanding the disease. A hint towards pathogenesis may be found in prior work in conjunction with our new findings and involves an unusual susceptibility of motor neurons to various forms of cell stress, particularly ER stress.

Accordingly, the membralin KO mouse generated in our laboratory provides a good model to study selective motor neuron vulnerability. These mice display severe motor neuron loss and muscle wasting, leading to paresis and death. Although the early disease onset in membralin KO mice is similar to that seen in an SMA mouse model (*Hsieh-Li et al., 2000*; *Monani et al., 2000*), the pattern of motor neuron injury in membralin KO mice is reminiscent of human distal HMN rather than proximal HMN, as observed in SMA. Of the dozen known causal genes for distal HMN, only a few show autosomal recessive inheritance, and no mouse model has, heretofore, been generated (*Irobi et al., 2004*, *2006*; *Irobi-Devolder, 2008*). Therefore, the membralin null mouse represents a novel model for human distal HMN. Additionally, our studies demonstrate the previously unknown function of membralin in motor neuron survival. Our findings from endogenous and heterologous expression studies suggest that membralin is mainly located in the ER. Relatively low expression of membralin seems sufficient for normal function, as heterozygotes of the membralin KO survived, and transgene rescue of the membralin KO did not require high expression levels. Importantly, in our rescue experiments, membralin transgene expression was primarily in brain, not muscle, consistent with the notion that the pathogenic process is predominantly neural in origin. The exact mechanisms underlying motor neuron death in membralin KO mice are not yet clear, although death occurs very rapidly at an early and well-defined postnatal stage. Thus, membralin KO mice can be used not only as an early-onset model of motor neuron disease but also to determine if there are features in common with late-onset motor neuron disease, such as motor neuron-specific vulnerability to ER stress (*Saxena et al., 2009*; *Roselli and Caroni, 2015*), dying back axonopathy (*Fischer et al., 2004*; *Coleman, 2005*), or non-cell autonomous death (*Boillee et al., 2006*; *Yamanaka et al., 2008*; *Kang et al., 2013*).

Previously, membralin was predicted to encode a transmembrane protein, but it lacked homology with any known protein domains and was not known to interact with other membrane proteins (*Andersson and von Euler, 2002*). In the present study, we found that membralin interacts with Erlin2, an ER membrane protein potentially involved in ERAD (*Christianson et al., 2012*). The exact cellular functions of Erlin2 are not clear, although it has been reported to interact with several ER resident E3 ligases such as GP78 and Hrd1 that are important for ERAD (*Christianson et al., 2012*). Additionally, Erlin2 has been shown to regulate ER membrane proteins such as Inositol 1,4,5-trisphosphate receptors (*Pearce et al., 2007*). It is conceivable that membralin assists Erlin2 in a complex that retrotranslocates unfolded proteins from the ER lumen to the cytosol, thus facilitating their ubiquitination for degradation (*Schulze et al., 2005*; *Carvalho et al., 2006*; *Denic et al., 2006*; *Vembar and Brodsky, 2008*; *Carvalho et al., 2010*; *Smith et al., 2011*; *Brodsky, 2012*). Consequently, membralin deficiency might increase both basal ER stress and vulnerability to ER stress-induced cell death, and this is exactly what we observed. We also found differences in Isg15 levels in membralin KO mice, leading to ISGylation, a process that represents a type I interferon-dependent, ER stress-triggered event. This observation is interesting in light of prior findings showing that ISGylation is a pre-symptomatic event observed in a mouse model of ALS (*Wang et al., 2011*). Collectively, our data suggest that the loss of membralin may contribute to degeneration by increasing ER stress in motor neurons, which are especially vulnerable to such stress due to their large metabolic demand. This demand is particularly prevalent postnatally when maturation of motor neurons requires synthesis of proteins for production of dendritic trees, synaptic connections, and ionic conductances (*Vinay et al., 2000*; *Carrascal et al., 2005*; *Li et al., 2005*).

Additionally, disruption of ERAD in SOD1 mutant mice is known to induce ER stress, activation of ASK1, and motor neuron cell death (*Nishitoh et al., 2008*). Mutations in Erlin2 have been linked to human disease, including intellectual disability, motor dysfunction, hereditary spastic paraplegia, and juvenile primary lateral sclerosis (*Al-Saif et al., 2012*; *Al-Yahyaee et al., 2006*; *Alazami et al., 2011*; *Wakil et al., 2013*; *Yildirim et al., 2011*). Moreover, disturbance in various other components of ERAD, including OS-9, erasin, ubiquilin2, torsinA, and Derlin1, causes ER stress (*Nishitoh et al., 2008*; *Alcock and Swanton, 2009*; *Lim et al., 2009*; *Deng et al., 2011*; *Nery et al., 2011*), and some of these genes have been linked to ALS (*Nishitoh et al., 2008*; *Alcock and Swanton, 2009*; *Lim et al., 2009*; *Alazami et al., 2011*; *Nery et al., 2011*; *Yildirim et al., 2011*; *Al-Saif et al., 2012*; *Wakil et al., 2013*). Therefore, future elucidation of the exact role of membralin in ERAD will undoubtedly be important for understanding the contribution of ER stress to motor neuron diseases. Interestingly, there are more than 50 single nucleotide polymorphisms (SNPs) in the coding region of human membralin that result in missense mutations (NBCI SNP database). The minor allele frequency of most SNPs is either low or has not been determined, suggesting that these SNPs are not common in the human population. Thus, our studies point to a specific role of membralin in motor neuron survival and potentially open a new avenue for research in the field of human motor neuron diseases.

## Materials and methods

### Constructing a target vector

We used the 3.6 kb (Apa I/Spe I) and 3.2 kb (BamH I/Kpn I) DNA fragments flanking the target region as the 5′- and 3′-arms, respectively, for the ~11.4 kb pGTN29/GluN3B/membralin targeting vector (see *Figure 1—figure supplement 1A*). Homologous recombination generated a ~5 kb size difference between the original and recombinant genes, facilitating their analysis. During construction of the targeting vector, one SpeI site at the junction of the 5′-arm and the target region were eliminated, and another SpeI site was introduced at the junction of the target region and 3′-arm. These restriction site variations allowed us to designed a 3′ probe, a 436 bp fragment from Xho I digestion, that hybridized to the 14 kb and 6 kb fragments generated by SpeI digestion or 9.1 kb and 8.4 kb fragments generated by EcoR I digestion (see *Figure 1—figure supplement 1A,B*) from the WT and recombinant alleles, respectively.

### Generation of an ES cell line and GluN3B/membralin DKO mice

All described procedures for animal were approved by the Institutional Animal Care and Use Committee of Sanford–Burnham Medical Research Institute and conducted in compliance with the Guide for the Care and Use of Laboratory Animals. Both sexes of mice were used for experiments and maintained in an institute facility accredited by the Association for Assessment and Accreditation of Laboratory Animal Care (AAALAC). The C1 ES cells were cultured in the presence of leukemia inhibitor factor on primary embryo fibroblast feeder cells and were transfected with the linearized pGTN29/GluN3B/membralin target vector DNA by electroporation at 400 V, 25 mF. Thirty-six hours after transfection, 0.48 mg/ml G418 was added to select for cells that have acquired the Neo$^r$ gene by homologous recombination with the targeting vector. The ES cells with the correct recombination were confirmed by standard Southern blot analysis using the probe described above. GluN3B/membralin-targeted ES cell lines were used for blastocyst injection. Then, 6 to 10 embryos, including 2 uninjected blastocysts as carriers, were transferred into one uterine horn of each pseudopregnant foster mother (F1 of DBA X C57BL/J6j). Chimeric mice were identified by eye pigmentation. Further breeding was made to test for germ line transmission of the injected ES cells. Mouse tail DNA was used to identify homozygotes and heterozygotes by the same procedure used in testing ES cells. Once the DKO line was established, we also designed a PCR method to simplify genotyping using a pair of primers for the GluN3B sequence in WT alleles and a pair of primers for the Neo sequence in the DKO allele (see *Supplementary file 1*).

### Generation of transgenic mouse lines carrying GluN3B and membralin transgenes

We have obtained the mouse full-length GluN3B cDNA clone from Dr Yuzaki as a gift (*Matsuda et al., 2002*). A full-length mouse membralin cDNA clone (ID# 3813678) was purchased from distributors of the Integrated Molecular Analysis of Genomes and their Expression (I.M.A.G.E)

consortium (http://www.imageconsortium.org/). We used the mouse GluN3B promoter that drives gene expression mainly in motor neurons from late embryonic stages to adult (database from NINDS Gensat Bac Transgenic Project, http://www.gensat.org/index.html). In addition, we used the mouse prion promoter, which drives high transgene expression in mouse neurons (*Baybutt and Manson, 1997*; *Loftus et al., 2002*; *Gispert et al., 2003*), to express the transgene in a broader brain area. The selected full-length transgene was subcloned downstream of either promoter, followed by an IRES or a sequence coding the GFP. The insert containing promoter, transgene, and IRES/GFP sequences were isolated by restriction digest followed by gel purification. The purified DNA fragments were injected into fertilized eggs to generate transgenic mice (C57BL/J6j) in our institute's Transgenic Facility according to established protocol. We used PCR methods to genotype transgenic mice using a pair of primers (see *Supplementary file 1*).

### Rescue of DKO mice by GluN3B and membralin transgenes

Hemizygotes of transgenic mice were bred with DKO mice to obtain double heterozygotes [transgene$^{+/-}$/ (GluN3B/membralin)$^{+/-}$]. The double heterozygotes were further bred to obtain mice that express the transgene in the GluN3B/membralin null background [transgene$^{+/-}$/(GluN3B/membralin)$^{-/-}$], and tested for GFP and transgene expression using RT-PCR. The crossed transgene/DKO mouse line was subjected to histological tests for examining motor neuron integrity. The survival time of the [transgene$^{+/-}$/(GluN3B/membralin)$^{-/-}$] mice were monitored and compared with that of DKO mice to determine the ability of the transgene to rescue the DKO mice phenotype.

### Generation of membralin KO mice by a gene trapping method

We generated membralin KO mice by using ES cells (gift of Sanger Institute) with a disrupted membralin gene by a gene trapping method (*Brennan and Skarnes, 1999*; *Skarnes et al., 2004*). Briefly, a trapping vector containing an RNA splicing acceptor sequence was inserted between exon 1 and 2 of the membralin gene in ES cells to disrupt normal RNA splicing. The positive ES cell clone was confirmed by sequencing and used for blastocyst injection to generate membralin KO mice (C57BL/J6j). RT-PCR experiments using primers from exon 1 and 2 further confirmed that homozygote mice carrying membralin/trapping vector alleles did not express membralin mRNAs with sequences beyond exon 2.

### Immunostaining and histological analysis

Previously established protocols were used for staining spinal cord sections or cultured cells (*Xing et al., 2006*), with the following antibodies: mouse anti-GFAP (Sigma-Aldrich, St. Louis, MO), rabbit anti-IBA-1 (Wako Chemicals, Inc., Richmond, VA), and anti-Hb9 (a gift from Dr Pfaff), followed by incubation with either a fluorophore-attached (Life Technologies, Grand Island, NY) or a biotinylated secondary antibody (Vector Laboratories, Burlingame, CA). Immunosignals were detected either directly under epifluorescence microscopy or after using a Vectastain Elite ABC kit (Vector Laboratories) with 3,3′-diaminobenzidine visualization (Roche Applied Science, Indianapolis, IN). The number of motor neurons in layers VIII and IX of the lumbar spinal cord was counted using stereological methods in WT, DKO, and membralin KO mice (*Coggeshall and Lekan, 1996*). Briefly, consecutive sections (12 μm in thickness) of the lumber enlargement (L1–L5) were collected for immunostaining with anti-Hb9 antibody to identify motor neurons. The number of motor neurons in each section was counted stereologically using adjacent sections for reference and look-up (physical disector). The total number of motor neurons was obtained for each mouse, and the mean value for 3 mice at each age group was calculated and normalized to that of WT mice. The general appearance of motor neurons was examined by conventional staining methods used in analysis of SOD1 mutant mice, including hematoxylin/eosin and cresyl violet staining.

HEK 293 cells were used for investigating the subcellular localization of membralin. A membralin-myc fusion protein was constructed by tagging the C-terminal of membralin with a Myc sequence using the pcDNA3.1 (−) Myc/his vector (Life Technologies). HEK 293 cells were plated onto coverslips, transfected with the membralin-Myc overnight, and fixed 48 hr after transfection with 4% PFA with 0.5% Triton X-100 in PBS. After blocking with 10% normal goat serum in PBS for 60 min, the cells were double stained by anti-Myc rabbit polyclonal antibody (Sigma) for membralin-Myc detection, followed by Alexa fluor-conjugated secondary antibodies (Life Technologies). Subcellular organelles were stained by either anti-PDI antibody (Enzo Life Science, Farmingdale, NY) for ER labeling, anti-GMP130

antibody (BD bioscience, Franklin Lakes, NJ) for Golgi labeling, or anti-mitochondria antibody (113-1, Abcam, Cambridge, MA) for mitochondrial labeling, followed by Alexa fluor-conjugated secondary antibodies (Life Technologies). DAPI was used to stain nuclei. Images were captured under confocal microscopy (Zeiss 710).

## Electron microscopy

Dorsal and ventral roots of lumbar segments (L3–L5) of the spinal cord were dissected out from P5 WT and DKO mice. Samples were immediately placed in 4% paraformaldehyde plus 1% glutaraldehyde in 0.1 M phosphate buffer and simultaneously processed for electron microscopy as described previously (*Liu et al., 1998*). Briefly, samples were osmicated in 1% $OsO_4$ for 10–20 min followed by washing in 0.1 M phosphate buffer and then dehydrated in graded ethanol and 100% acetone. Each sample was oriented and placed in a Flat Embedding Mold (Ted Pella, Redding, CA) filled with Araldite (EMS, Fort Washington, PA). Embedded dorsal or ventral roots of different genotypes were dissected, and the cross sections of the roots were re-cut on an ultramicrotome (Ultracut, Leica) at 70–80 nm. Serial thin sections were collected on Formva-coated single slot nickel grids and stained with uranyl acetate and lead citrate. Ultrathin sections were examined in a Philips CM120 electron microscope at 80 KV. Digitized images were acquired by a high-resolution (2K × 2K) CCD camera (Gatan, Inc., Pleasanton, CA), processed using software provided by the manufacturer (DigitalMicrograph), and displayed with Photoshop CS (Adobe Systems, San Jose, CA).

## Northern blot analysis

Two DNA fragments were purified by enzymatic digestion of membralin cDNA using BamH I/Bgl II and Not I/BamH I, respectively, to generate two probes specific to regions of exon VII-X and the last exon (XI) of membralin. Northern blot analysis was performed using these two probes labeled with $^{32}$P by random priming. 10 µg of total mouse RNA per lane were analyzed by electrophoresis on a 1.1% denaturing gel and subsequently transferred to a nylon membrane. The blot was hybridized at 42°C in a solution containing 50% formamide, 6x SSC, 5x Denhardt's reagent, and 0.5% SDS. The blots were dried and exposed to autoradiographic films for analysis.

## Analysis of the interaction between membralin and Erlin2

cDNAs for mouse membralin and Erlin2 were purchased from distributors of the Integrated Molecular Analysis of Genomes and their Expression (I.M.A.G.E) consortium (http://www.imageconsortium.org/) and subcloned into a pcDNA3.1 vector for mammalian cell expression. C-terminal tagged membralin (membralin-myc or membralin-mCherry) and N-terminal or C-terminal tagged Erlin2 (Erlin2-EGFP or Erlin2-HA) were constructed by introducing corresponding tag fragments generated by the PCR method, and all constructs were confirmed by sequencing. Membralin-myc and Erlin2-HA were co-transfected in HEK 293 cells and total cell lysates collected in 48 hr after transfection for co-immunoprecipitation-immunoblot assay.

The interaction of Erlin2-EGFP and membralin-mCherry was validated by the acceptor photobleaching method for FRET detection (*Karpova and McNally, 2006*). Briefly, HEK 293 cells were transfected with Erlin2-EGFP and membralin-mCherry for 24 hr. A Zeiss 710 NLO microscope (Carl Zeiss Inc.) was used to record the fluorescence of EGFP and mCherry in transfected cells. Three pre-bleached and five post-bleached images were acquired. Averaged fluorescence intensities of the donor were calculated from the measurement of regions of interest for each experimental set before and after bleaching. The efficiency of FRET was calculated by $E_{fret} = 1 - (I_a/I_b)$, where $I_a$ and $I_b$ represent the steady-state donor fluorescence in the presence and the absence of the acceptor, respectively. Seven Ala-linked EGFP-mCherry plasmids were transfected into HEK 293 cells and were used as a positive FRET control (FRET efficiency = ~50%, data not shown). FRET efficiencies were reported as mean ± s.e.m.

## Assays for detecting alterations in ER stress signaling pathways

We measured levels of GRP78, CHOP, and ATF4, components of three canonical branches of UPR, by Western blot analysis using the spinal cord tissues and MEFs from membralin KO mice and WT littermates. The antibodies against these proteins were purchased from commercial sources: GRP78 (H-129, 1:1000; Santa Cruz Biotechnology, Santa Cruz, CA), CHOP (2895S, 1:1000; Cell Signaling, Denvers, MA), and ATF4 (SC200, 1:500; Santa Cruz Biotechnology). Total RNA was

isolated from the spinal cord of membralin mutant mice and WT littermate using TRIzol reagent (Life Technologies). The following set of primers was used to detect the expression of mouse Xbp-1 (GATCCTGACGAGGGTCCAAGA and ACAGGGTCCAACTTGTCCAG).

## Toxicity assay in vitro

MEFs from membralin KO mice and WT littermates were isolated from day 12.5 embryos and cultured by conventional methods. In the toxicity assay, the percentage of cell death was measured by counting cells or using the CellTiter 96 Aqueous Non-Radioactive Cell Proliferation Assay kit (Promega, Madison, WI) after MEF cells were exposed to the ER stress inducer, tunicamycin, for 24 hr.

## Statistical analysis

The sample size was 3–10 per genotype for animal histology and 3–6 for protein or transcript expression as well as for cell viability studies, as determined by Power Analyses of previous data. The experiments were not randomized. Although the initial investigator performing the experiments was not blinded, the samples and animal results were then examined by a group of the investigators masked to experimental identity in order to evaluate the results. All data points demonstrated a normal distribution and were all included in the analysis. Data are presented as mean ± s.e.m. and analyzed by a Student's $t$-test for pairwise comparisons. Statistical analyses were conducted using GraphPad Prism software (version 6). A p value <0.05 was considered statistically significant.

## Acknowledgements

We thank M Yuzaki at Keio University for the mouse full-length GluN3B cDNA clone and A Aguzzi at University Hospital of Zurich for the mouse prion promoter. We thank Wellcome Trust Sanger Institute for providing ES cells carrying membralin deletion mutant gene by trapping method. We thank the animal facility at the Sanford–Burnham Medical Research institute for generating and maintaining mutant mouse lines. This work was supported in part by NIH grants to SAL (P01 HD29587, R01 NS086890, and P30 NS076411) and DZ (R01 NS043434).

## Additional information

### Funding

| Funder | Grant reference | Author |
|---|---|---|
| National Institutes of Health (NIH) | R01 NS43434 | Dongxian Zhang |
| National Institutes of Health (NIH) | P01 HD29587 | Stuart A Lipton |
| National Institutes of Health (NIH) | P30 NS076411 | Stuart A Lipton |
| National Institutes of Health (NIH) | R01 NS086890 | Stuart A Lipton |

The funder had no role in study design, data collection and interpretation, or the decision to submit the work for publication.

### Author contributions

BY, RW, JEC, DZ, Conception and design, Acquisition of data, Analysis and interpretation of data, Drafting or revising the article; MQ, SN, Conception and design, Acquisition of data, Analysis and interpretation of data; X-BL, BZ, Acquisition of data, Analysis and interpretation of data, Drafting or revising the article; JLM, NN, SAL, Conception and design, Analysis and interpretation of data, Drafting or revising the article; KS, Analysis and interpretation of data, Drafting or revising the article, Contributed unpublished essential data or reagents

### Ethics

Animal experimentation: All described procedures for animal were approved by the Institutional Animal Care and Use Committee of Sanford–Burnham Medical Research Institute and conducted in compliance with the Guide for the Care and Use of Laboratory Animals (Animal Use Form #14-060). Both sexes of mice were used for experiments and maintained in an institute facility accredited by the Association for Assessment and Accreditation of Laboratory Animal Care (AAALAC).

# Additional files

**Supplementary file**
• Supplementary file 1. Primer list for genotyping and sequencing.

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
