## [Decision Letter]

Thank you for sending your work entitled “The critical role of membralin in postnatal motor neuron survival and disease” for consideration at eLife. Your article has been favorably evaluated by a Senior editor and three reviewers, one of whom is a member of our Board of Reviewing Editors.

The Reviewing editor and the other reviewers discussed their comments before we reached this decision, and the Reviewing editor has assembled the following comments to help you prepare a revised submission.

This manuscript describes a potential function for a novel protein, membralin, in the ER stress response and survival of neonatal motor neurons. These findings have clinical significance given the dramatic motor neuron degeneration phenotype seen in the membralin KOs, and the finding that the membralin-binding partner Erlin2 is implicated in human motor neuron disorders. In general, the study is well-performed, and the conclusions are intriguing. In addition, the work describes an interesting function for a previously poorly-characterized protein. However, the reviewers felt that additional data was required to strengthen the major conclusions of the paper, as follows.

1) One of the key conclusions of the manuscript is that membralin knockout causes motor neuron death. However, while there is nice evidence for functional loss of motor neurons and motor axon degeneration, the evidence for motor neuron cell death is not completely convincing. The tissue sections shown in Figures 1 and 3 do not match between wildtype and knockouts with respect to anatomical position, and it would be better if the authors counted motor neurons from serial sections in a specific population of neurons, such as L1-L5 or facial nucleus. It would also be helpful if more details about the methods of quantification were provided.

2) The cell biological data with membralin should be strengthened. The data on colocalization, shown in Figure 4, are not convincing as presented. This may be because the images are so low-magnification. In addition, it would greatly strengthen the paper to show that endogenous Erlin2 and membralin interact either in the brain or in the MEFs studied for much of the cell biology here. While the membralin antibodies might not work for immunoprecipitations, could the authors use Erlin2 antibodies for this experiment, since the membralin antibodies work well on westerns? Finally, the co-immunoprecipitation experiments require controls and more of the western blots should be shown.

3) The number of animals in the rescue experiment seems very low. Could the authors clarify how many animals were used in this experiment, and how many independent litters they came from?

4) Protein data confirming the loss of membralin in the two mouse models is important, particularly given the nature of the construct used in the double knockout. This appears to be shown for the single knockout in Figure 3, bottom panel, but the figure legend doesn't indicate this. Is that the case? A similar western blot should also be shown for the double knockout.

5) Figure 6 shows upregulation of GRP78 and ATF4 in P3 spinal cord. It would be interesting to see whether this upregulation is also observed at earlier stages, before any motor defect is observed, in order to correlate the upregulation with the onset of disease symptoms.

---

## [Author Response]

*1) One of the key conclusions of the manuscript is that membralin knockout causes motor neuron death. However, while there is nice evidence for functional loss of motor neurons and motor axon degeneration, the evidence for motor neuron cell death is not completely convincing. The tissue sections shown in*
Figures 1 and 3
*do not match between wildtype and knockouts with respect to anatomical position, and it would be better if the authors counted motor neurons from serial sections in a specific population of neurons, such as L1-L5 or facial nucleus. It would also be helpful if more details about the methods of quantification were provided*.

In the revised manuscript, we use a stereological method (19) to determine the number of motor neurons in layers VIII and IX of the lumbar spinal cord in WT, DKO, and membralin KO mice. Briefly, consecutive sections (12 µm in thickness) of the lumber enlargement (L1-L5) were collected for immunostaining with anti-Hb9 antibody to identify motor neurons. The number of motor neurons in each section was counted based on stereological counting methods using adjacent sections as a reference and look-up section, respectively (physical disector). We have added these details to the Methods section (in the subsection headed “Immunostaining and histological analysis”). We believe this counting method represents the best estimation in determining the average number of motor neurons in the spinal cord of each genetic group. We found that matching exactly the anatomical location in the brain or spinal cord tissue of WT and KO mice at postnatal day 5 was not achievable due to the variation in size and fragility of the tissue (which lacks fully developed white matter).

*2) The cell biological data with membralin should be strengthened. The data on colocalization, shown in*
Figure 4*, are not convincing as presented. This may be because the images are so low-magnification. In addition, it would greatly strengthen the paper to show that endogenous Erlin2 and membralin interact either in the brain or in the MEFs studied for much of the cell biology here. While the membralin antibodies might not work for immunoprecipitations, could the authors use Erlin2 antibodies for this experiment, since the membralin antibodies work well on westerns? Finally, the co-immunoprecipitation experiments require controls and more of the western blots should be shown.*

We have now repeated co-localization experiments with some modification such as using an improved mitochondrial marker. Also, images of higher resolution were taken by confocal microscopy. These results are presented in the revised manuscript (Figure 4).

We have also attempted the co-IP experiment three times to demonstrate the interaction between endogenous membralin and Erlin2. However, these experiments were not successful due to either the low quality of the antibody or very low concentration of the endogenous proteins. In the reviewed manuscript, we have improved our co-IP experiments by adding more controls and by using anti-Erlin2 antibody directly (replacing anti-HA antibody), so we are more confident that the detected bands indeed represent Erlin2 protein.

*3) The number of animals in the rescue experiment seems very low*. *Could the authors clarify how many animals were used in this experiment, and how many independent litters they came from*?

The number of animals shown in Figure 2 is an example of data obtained from all littermates in the same litter. We have observed a consistent rescue effect in 5 independent litters (representing breeding among independent heterozygous DKO mice positive for the membralin transgene). We have clarified this procedure in the revised text (Figure 2 legend). We bred homozygous DKO mice with the membralin transgene rather than heterozygous because it reduces the number of mice used and conserves funds.

*4) Protein data confirming the loss of membralin in the two mouse models is important, particularly given the nature of the construct used in the double knockout. This appears to be shown for the single knockout in*
Figure 3*, bottom panel, but the figure legend doesn't indicate this. Is that the case? A similar western blot should also be shown for the double knockout*.

We chose Southern and Northern blots for the analysis of membralin deletion in double KO (DKO) mice, as knockout membralin protein has a similar size as WT membralin, so the two cannot be distinguished on a western blot. In addition, it is extremely difficult to detect the endogenous protein due to either the low amount of endogenous membralin or the poor quality of the available antibody. The membralin protein data shown in Figure 3 were obtained by using the membrane fraction of protein purified from approximately half of the brain (this is now indicated in the figure legend). We currently do not have the DKO mice to perform additional western blots and cannot obtain additional animals in time to meet the deadline imposed by the Editor for resubmission.

*5)*
Figure 6
*shows upregulation of GRP78 and ATF4 in P3 spinal cord. It would be interesting to see whether this upregulation is also observed at earlier stages, before any motor defect is observed, in order to correlate the upregulation with the onset of disease symptoms*.

Since membralin KO mice at P3 showed no behavioral abnormality or motor neuron loss, we consider P3 a pre-symptomatic stage. We agree with the reviewers that it would be interesting to see if upregulation of ER stress markers occur at an even earlier stage and we are currently conducting these experiments using embryos and P0-P2 mice. However, we feel that these studies are for future consideration.